

# Investigating the "off-hour effect" on outcomes of neonates undergoing emergency gastrointestinal surgery

Yu Cui

The Affiliated Hospital, School of Medicine, UESTC Chengdu Women's & Children's Central Hospital, Chengdu, Sichuan, China

## ABSTRACT

**Background**. Evidence regarding the off-hour effect on outcomes in neonates with gastrointestinal disease who received urgent surgical intervention is unknown. Because of the workforce shortage, insufficient experience of surgeons, and delayed radiography examination during off-hours, we hypothesized that emergency surgeries performed during off-hours were associated with worse outcomes. This study aims to analyze the association between the "off-hour effect" and adverse events of neonates undergoing emergency gastrointestinal surgery.

**Methods**. We extracted patient data from the electronic medical record system at our institution for all neonates undergoing emergency gastrointestinal surgery between July 2018 and October 2021. The primary outcomes were 24-hour and in-hospital mortality. The secondary outcomes were actual postoperative length of stay (PLOS) and the incidence of unplanned re-operation.

**Results**. A total of 275 neonates were identified, and 207 (75.3%) were treated during off-hours. The "off-hour effect" was not associated with increased 24-hour mortality, in-hospital mortality, PLOS, and unplanned re-operation. After propensity score matching, 68 off-hours were matched to the nearest 68 on-hours based on their age, weight, gestation weeks, and American Society of Anesthesiologists (ASA) status. No differences were detected in the primary and secondary outcomes.

**Conclusion**. In this retrospective study with neonates who underwent emergency gastrointestinal surgery, after controlling for age, weight, gestation weeks, and ASA status, surgical and medical outcomes were not different in those undergoing off-hours surgery, which can be considered a surrogate for similar quality of care. However, in the future, a multi-center, prospective study is needed to confirm the results, to overcome the bias related to the presence of only one surgical team.

## INTRODUCTION

Symptoms suggestive of gastrointestinal disorders are the most common reason for neonates to undergo emergency surgery, such as neonatal necrotizing enterocolitis (NEC), intestinal obstruction, and gastric or intestinal perforation. These disorders may rapidly get worse and require immediate treatment, even during off-hours. Previous studies have summarized that NEC and sepsis are independent risk factors associated with

Corresponding author
Yu Cui, cuiyu19831001@163.com

perioperative complications in neonates (*Lillehei, Gauvreau & Jenkins, 2012*; *Michelet et al., 2017*). A 10-year investigation reported that 4.9% of newborns had acute abdomen, and early recognition and timely surgical intervention are the effective strategies to avoid deterioration and loss of functioning bowel (*Rocha et al., 2009*). According to our previous work, emergency gastrointestinal surgery for neonates accounts for over 70% (*Cui et al., 2021*). Undoubtedly, some patients had to accept surgical intervention during off-hours.

The quality of medical care in the different periods has been well studied. In 2001, a retrospective study including more than 3 million cases proposed critical patients were more likely to die in the hospital if they were admitted on a weekend than if they were admitted on a weekday (*Bell & Redelmeier, 2001*). Subsequently, *Ayar et al. (2019)* presented that the risk of mortality was higher for pediatrics in an intensive care unit during off-hours than that during on-hours. Neonates delivered on off-hours tended to have a higher proportion of developing neonatal respiratory distress syndrome and intrauterine pneumonia (*Wang et al., 2021*). Moreover, labor on weekends, during the night shift was associated with significantly elevated risk for cerebral palsy (*Toyokawa et al., 2020*). This phenomenon is called the "off-hour effect". However, several studies refuted these viewpoints, and some clinicians had proved that no clinical differences were detected in outcomes among the different presenting hours. A study about the association between off-hour admission in critical children reported that off-hour admission did not relate to mortality (*Kido et al., 2021*). Besides, extubating children during off-hour did not have any unfavorable outcomes (*Da Silva et al., 2016*). Data from 572 premature infants indicated that despite potential fluctuations in staffing levels during on- and off-duty hours, morbidity and mortality of infants were not affected (*Akin & Cakir, 2024*). Similar meta-analyses enrolled in 10 studies proved that the time of pediatric intensive care unit admission did not have significant differences in the odds of mortality (*Williams et al., 2020*).

However, evidence regarding the off-hour effect on mortality in neonates with the gastrointestinal disease who received urgent surgical intervention is unknown. Because of the workforce shortage, insufficient level of surgeons' experience, and delayed radiography examination during off-hours, we hypothesized that emergency surgery during off-hours was associated with worse outcomes. We, therefore, performed a retrospective study to evaluate the association between off-hours emergency surgical intervention and hospital mortality in neonates with gastrointestinal disorders.

## MATERIALS AND METHODS

### Study design and setting

A single-center retrospective cohort study was designed to analyze the association between the "off-hour effect" and adverse events of neonates undergoing emergency gastrointestinal surgery. The study was approved by the Institutional Review Board (or Ethics Committee) of Chengdu Women's and Children's Central Hospital (No. B2021(27)) on 10/25/2021. We obtained patient data from the electronic medical record system at our institution for all neonates undergoing emergency gastrointestinal surgery between July 2018 and October 2021. The informed written consent was waived due to the anonymous patients' data. This

study was reported according to the STROCSS (Strengthening the Reporting of Cohort Studies in Surgery) criteria.

## Patient selection

All neonates (aged≤28 days) undergoing gastrointestinal surgery were enrolled. Subjects were excluded if they met the exclusion criteria as follows: ① The patients underwent elective surgery; ② they did not have sufficient data to be analyzed. Patients were divided into two cohorts depending on the period of surgical intervention (off-hour group and on-hour group). An off-hour group was defined as an operation started between 5:01 PM–7:59 AM on a weekday or any time on a weekend day or a statutory holiday, otherwise, the patients were categorized into an on-hour group.

In our institution, all newborns with the potential possibility of surgical intervention are consulted by the pediatric surgeon. During on-hours, the senior or junior attending physician evaluates the patient and initiates surgical intervention, while during off-hours, only the junior attending physician makes a judgment and provides surgical intervention. Senior attending physicians were defined as physicians with ≥3 years of experience in pediatric surgery, while junior attending physicians were defined as physicians with <3 years of experience in pediatric surgery. In addition, two anesthesiologists (at least one of them with special training in pediatric anesthesia) and two nurses (experienced in pediatric care) provide surgical service for a neonate in the operating room. If two operations require starting, the backup team members will be informed, and they should appear within 30 min. Neonates were transferred to either the neonatal intensive care unit (NICU) or the Surgical Intensive Care Unit (SICU) after surgery, depending on the patient's admission department before surgery. In our hospital, SICU is a branch of the pediatric intensive care unit (PICU), which provides medical care for pediatric patients after surgery.

## Clinical data collection

The demographic data were collected from the electronic medical record system, including gender, days, weight at surgery, birth weight, gestational weeks, American Society of Anesthesiologists (ASA) status, preoperative hemoglobin, and diagnosis. Intraoperative information was also extracted, *i.e.,* the length of surgical time, duration of anesthesia, mean body temperature, estimated blood loss, blood transfusion, fluid infusion, intraoperative urinary output, intraoperative vasopressor support, and the use of furosemide. Postoperative variables were postoperative hemoglobin, postoperative length of stay (PLOS), unplanned re-operation, 24-h mortality, and hospital mortality. The body temperature was recorded at 5 min intervals, and the mean body temperature was defined as the average temperature during the surgical period. The administration of one or more of the following medications (*i.e.,* dobutamine, dopamine, norepinephrine, epinephrine, vasopressin, and phenylephrine) during the intraoperative period was considered having intraoperative vasopressor support, which was in line with our previous study (*Cui, Cao & Deng, 2022*).

## Primary and secondary outcomes

The primary outcomes were 24-hour and in-hospital mortality. Mortality data were obtained from a combination of death records and hospital discharge disposition. Newborns were deemed dead if they were being mechanically ventilated, but were authorized to give up treatment by a legal guardian.

The secondary outcomes were actual PLOS and the incidence of unplanned re-operation. Actual PLOS was calculated from the surgical date to discharge. An unplanned re-operation was defined as an unexpected acceptance of surgery due to complications related to the original surgical intervention (*Cui et al., 2021*).

## Sample size calculation

A meta-analysis presented that off-hours hospital admission was associated with higher mortality in patients with upper gastrointestinal hemorrhage (*Xia et al., 2018*). Based on our previous work (*Cui et al., 2021*), about 200 newborns underwent gastrointestinal surgery per year and 70% of them underwent emergency surgical intervention. Of these, the mortality of patients who underwent emergency surgical intervention was about 13.4%. Assuming a 70% decrease in mortality (to 4%) to be clinically important and a sampling ratio of 2:1 (according to the length of the time period), we estimated that 204 patients would be required to detect a power of 0.8 and a significance level of 0.05. The group sample sizes were 136 in the off-hour group and 68 in the on-hour group, respectively.

## Statistical analysis

First, the distribution of continuous variables was detected by the Shapiro–Wilk test. Then, the continuous variables were presented as the mean ± standard deviation (SD) if normally distributed, otherwise reported as a median and interquartile range (IQR) (25–75%). The student $t$-test was used to compare normally distributed continuous parameters, otherwise, the Mann–Whitney U-test was used to compare two groups. The categorical variables were expressed as a percentage (%), and chi-square statistics or Fisher's exact test was used to find differences between groups as appropriate. Furthermore, when there were significant differences in basic characteristics between the two groups, a propensity score method (PSM) was conducted to make the patients in the off-hour group and those in the on-hour group comparable. Age, weight, gestationdays, ASA status, surgical year, and diagnosis were selected as the matching variables. $P < 0.05$ was considered statistically significant. All the statistical analyses were performed with R Studio 3.6.3.

## RESULTS

Over a three-year period, 497 neonates underwent gastrointestinal surgery from 7/1/2018 to 10/1/2021. Of them, 125 neonates did not have sufficient information. In addition, 97 patients who underwent elective surgery were excluded. Finally, 275 neonates were enrolled, and 207 (75.3%) of them were treated during off-hours. The study flow chart was shown in Fig. 1.

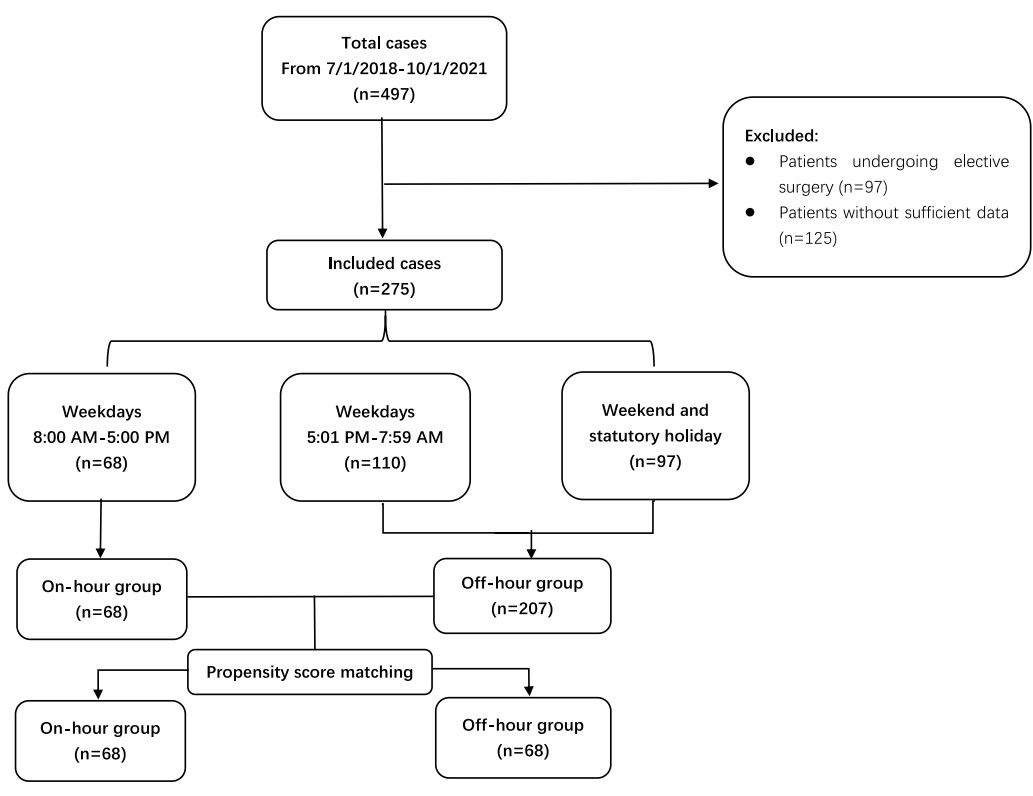

**Figure 1  The study flow chart.**

## Demographics and patient characteristics

The clinical demographics of the 275 newborns who underwent emergency gastrointestinal surgery were reported in Table 1. A total of 207 cases (75.3%) received surgical intervention during off-hours. The median (IQR) age of enrolled patients was 5.0 [3.0, 11.0] days. There were more males than females in both groups (43 (63.2%) *vs* 116 (56.0%); $P = 0.56$). The most frequent diagnosis was NEC (52.4%), followed by congenital anorectal malformation (16.4%), intestinal atresia (12.4%), and gastric perforation (4.4%). There were no differences in the surgical indication of the subjects between the two groups ($P = 0.12$). Among them, 16.0% of patients preoperatively needed a transfusion and the median (IQR) of preoperative hemoglobin was 144.0 (115.5, 170.5) g/L. At the time of surgery, the body weight of neonates in on-hour group was heavier than that in the off-hour group (On-hour group: 2.9 (2.4, 3.3) *vs.* Off-hour group: 2.6 (2.1, 3.2) kg; $P = 0.03$). A similar trend in the birth weight of neonates between the two groups was observed (On-hour group: 2.9 (2.4, 3.3) *vs.* Off-hour group: 2.6 (2.0, 3.2) kg; $P = 0.02$). The gestational weeks of patients in the off-hour group were slightly less than those in the on-hour group (On-hour group: 38.0 (36.0, 39.4) *vs.* Off-hour group: 36.9 (34.9, 39.0) weeks; $P = 0.049$). About 63% of neonates in the on-hour group had an ASA status of III or IV, whereas 81.6% were in the off-hour group ($P < 0.01$). Intraoperative data were comparable between the two groups, including

**Table 1 Preoperative characteristics of patients in the on-hour and off-hour groups.**

| Variables | Before propensity score matching | | | | After propensity score matching | | |
|---|---|---|---|---|---|---|---|
| | Total (n = 275) | On-hour (n = 68) | Off-hour (n = 207) | P value | On-hour (n = 68) | Off-hour (n = 68) | P value |
| Age, days (Median (IQR)) | 5.0 (3.0, 11.0) | 4.0 (3.0, 9.3) | 6.0 (3.5, 11.0) | 0.12[a] | 4.0 (3.0, 9.3) | 5.0 (3.0, 8.0) | 0.80[a] |
| Gender, male (%) | 159 (57.8) | 43 (63.2) | 116 (56.0) | 0.37[b] | 43 (63.2) | 42 (61.8) | 1.00[b] |
| Weight | | | | | | | |
|    At surgical time, kg (Median (IQR)) | 2.7 (2.1, 3.2) | 2.9 (2.4, 3.3) | 2.6 (2.1, 3.2) | 0.03[a,*] | 2.9 (2.4, 3.3) | 2.9 (2.5, 3.3) | 0.97[a] |
|    Birth weight, kg (Median (IQR)) | 2.7 (2.1, 3.2) | 2.9 (2.4, 3.3) | 2.7 (2.0, 3.2) | 0.02[a,*] | 2.9 (2.4, 3.3) | 2.9 (2.3, 3.4) | 0.85[a] |
|    Weight<1.5 kg, n (%) | 23 (8.3) | 4 (5.9) | 19 (9.2) | 0.46[c] | 4 (5.9) | 7 (10.3) | 0.53[c] |
| Gestational weeks, weeks (Median (IQR)) | 37.3 (35.0, 39.1) | 38.0 (36.0, 39.4) | 36.9 (34.9, 39.0) | 0.049[a,*] | 38.0 (35.9, 39.4) | 37.7 (35.7, 39.0) | 0.46[a] |
| Premature, n (%) | 126 (45.8) | 22 (32.4) | 104 (50.2) | 0.02[b,*] | 22 (32.4) | 26 (38.2) | 0.59[b] |
| ASA status, n (%) | | | | <0.01[c,*] | | | 0.47[c] |
|    I | 1 (0.4) | 1 (1.5) | 0 (0.0) | | 1 (1.5) | 0 (0.0) | |
|    II | 62 (22.5) | 24 (35.3) | 38 (18.4) | | 24 (35.3) | 21 (30.9) | |
|    III | 191 (69.5) | 35 (51.5) | 156 (75.4) | | 35 (51.5) | 42 (61.8) | |
|    IV | 21 (7.6) | 8 (11.8) | 13 (6.3) | | 8 (11.8) | 5 (7.4) | |
| Diagnosis, n (%) | | | | 0.12[c] | | | 0.44[c] |
|    Appendicitis | 4 (1.5) | 1 (1.5) | 3 (1.4) | | 1 (1.5) | 1 (1.5) | |
|    Congenital anorectal malformation | 45 (16.4) | 18 (26.5) | 27 (13.0) | | 18 (26.5) | 14 (20.6) | |
|    Diaphragmatic hernia | 2 (0.7) | 1 (1.5) | 1 (0.5) | | 1 (1.5) | 0 (0.0) | |
|    Gastric perforation | 12 (4.4) | 1 (1.5) | 11 (5.3) | | 1 (1.5) | 2 (2.9) | |
|    Hernia | 8 (2.9) | 2 (2.9) | 6 (2.9) | | 2 (2.9) | 3 (4.4) | |
|    Ileus | 10 (3.6) | 5 (7.4) | 5 (2.4) | | 5 (7.4) | 0 (0.0) | |
|    Intestinal atresia | 34 (12.4) | 7 (10.3) | 27 (13.0) | | 7 (10.3) | 13 (19.1) | |
|    Intestinal malrotation | 5 (1.8) | 1 (1.5) | 4 (1.9) | | 1 (1.5) | 2 (2.9) | |
|    Intestinal stenosis | 1 (0.4) | 0 (0.0) | 1 (0.5) | | 0 (0.0) | 0 (0.0) | |
|    Intestinal Volvulus | 3 (1.1) | 1 (1.5) | 2 (1.0) | | 1 (1.5) | 1 (1.5) | |
|    NEC | 144 (52.4) | 28 (41.2) | 116 (56.0) | | 28 (41.2) | 30 | |
|    Pyloric hypertrophy | 1 (0.4) | 1 (1.5) | 0 (0.0) | | 1 (1.5) | 0 (0.0) | |
|    Omphalocele | 6 (2.2) | 2 (2.9) | 4 (1.9) | | 2 (2.9) | 2 (2.9) | |
| Preoperative transfusion, n (%) | 44 (16.0) | 9 (13.2) | 35 (16.9) | 0.50[b] | 9 (13.2) | 7 (10.3) | 0.79[b] |
| Preoperative hemoglobin, g/dl (Median (IQR)) | 144.0 (115.5, 170.5) | 146.0 (111.5, 173.5) | 144.0 (117.5, 168.5) | 0.96[a] | 146.0 (111.5, 173.5) | 153.0 (118.5, 173.8) | 0.33[a] |

**Notes.**
*$P < 0.05$.
[a] P value on a Mann–Whitney U-test.
[b] P value on a chi-squared test.
[c] P value on a Fisher exact test.

duration of anesthesia, duration of surgery, the requirement of vasopressor support, the administration of furosemide, estimated blood loss, fluid infusion, and urinary output.

To adjust for differences in basic characteristics of neonates between the two groups and to minimize confounding, a propensity score matching was performed after which all

demographic characteristics among off-hour *versus* on-hour groups were evenly balanced (Table 1).

## Primary and secondary outcomes

The overall 24-hour mortality was 10.9%, ranging from 9.7% in the off-hour group to 14.7% in the on-hour group ($P = 0.35$). Of them, the main diagnosis was NEC (80.0%, 16/20), intestinal atresia (10%, 2/20), and intestinal malrotation (10%, 2/20) during the off-hours, which was in line with NEC (60.0%, 6/10), diaphragmatic hernia (10%, 1/10), gastric perforation (10%, 1/10), intestinal atresia (10%, 1/10) and intestinal malrotation (10%, 1/10) during daytime.

The overall in-hospital mortality was 13.5% (37/275), with 17.7% (12/68) in the on-hour group and 12.1% (25/207) in the off-hour group ($P = 0.34$). No significant differences were detected between the two groups. The most common diagnoses were NEC (70.3%, 26/37), intestinal atresia (13.5%, 5/37), and intestinal malrotation (8.1%, 3/37). Detailed information about the two groups was shown in Fig. 2.

The median actual PLOS in the on-hour group was 16.0 (IQR: 11.0, 25.5) days *versus* 18.0 (IQR: 13.5, 25.5) days in the off-hour group ($P = 0.18$). The incidence of unplanned re-operation was not significantly higher in the off-hour group (16.2% *vs* 12.6%, $P = 0.58$).

## Propensity score–matched comparison

A propensity score matching was used to compare the on-hour cohort and the off-hour cohort. Finally, 68 off-hours were matched to the nearest 68 on-hours based on their age, weight, gestation weeks, and ASA status (Table 1). As shown in Table 1, the groups were comparable to demonstrate no significant differences in age, weight including birth weight and weight at surgery, gestational weeks, and ASA status ($P > 0.05$). Besides, the intraoperative variables, such as the duration of anesthesia, the duration of surgery, vasopressor support, the administration of furosemide, estimated blood loss, transfusion, fluid infusion, and urinary output, were comparable between the two cohorts (Table 2). Table 3 summarized the outcomes between the two groups. After propensity score matching, no differences were detected between the on-hour group and the corresponding off-hour group on primary and secondary outcomes.

## DISCUSSION

In this retrospective study, the timing (on-hours and off-hours) of emergency gastrointestinal surgery for neonates did not affect the 24-hour and in-hospital mortality, as well as actual PLOS and the incidence of unplanned re-operation, which disputed our hypothesis that emergency surgery during off-hours might be associated with worse outcomes. Although a propensity score–matched comparison was performed, the results were not changed. Thus, we have demonstrated that the "off-hour effect" impacts neither the primary nor secondary outcome for neonates who underwent emergency gastrointestinal surgery.

Notably, several basic characteristics significantly differed between the two cohorts, which was inevitable in a retrospective study. The neonates who accepted off-hour surgery

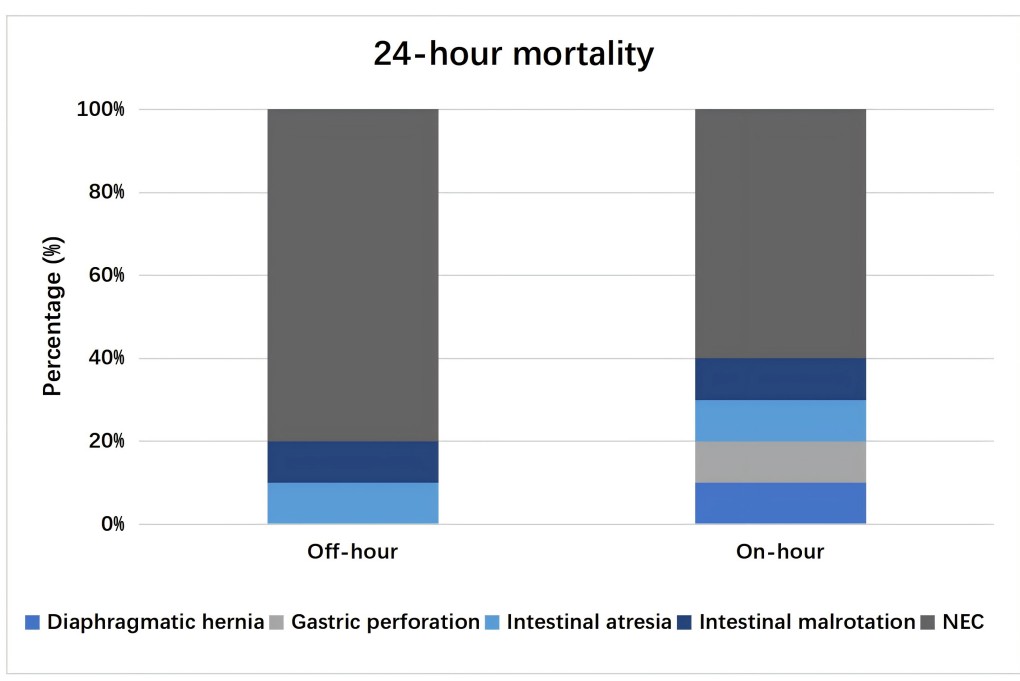

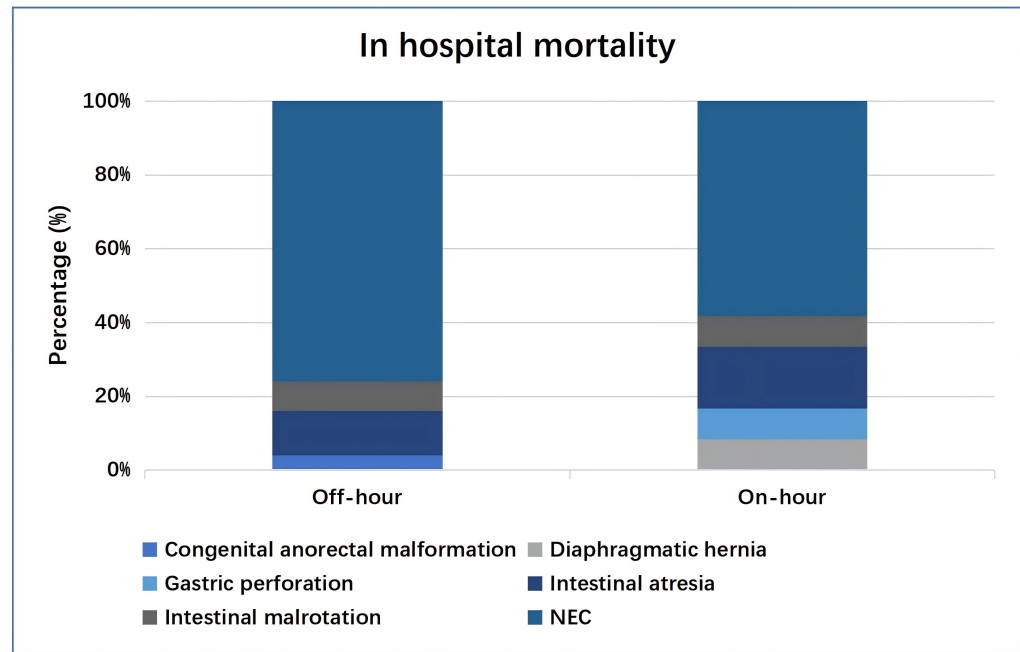

**Figure 2** Distribution of diagnosis of 24-hour death or in-hospital death.

had fewer gestational weeks and lower body weight (both birth weight and weight at surgery) than those during on-hours, suggesting that the cases in the off-hour group might be associated with more severe conditions. A previous study proved that both gestation age and being small for gestation were the risk factors to predict early neonatal death (*Abdel-Latif et al., 2006*). Thus, a propensity score match was performed to control

**Table 2 Intraoperative data.**

| Variables | Before propensity score matching | | | After propensity score matching | | |
|---|---|---|---|---|---|---|
| | On-hour (n = 68) | Off-hour (n = 207) | P value | On-hour (n = 68) | Off-hour (n = 68) | P value |
| Duration of anesthesia, min (Median (IQR)) | 161.0 (128.8, 205.0) | 170.0 (135.0, 205.0) | 0.61[a] | 161.0 (128.8, 205.0) | 155.0 (114.2, 195.8) | 0.47[a] |
| Duration of surgery, min (Median (IQR)) | 90.5 (64.0,126.0) | 100.0 (75.0, 122.0) | 0.61[a] | 90.5 (64.0, 126.0) | 91.5 (62.3, 116.3) | 0.52[a] |
| Vasopressor support, n (%) | 18 (26.5) | 65 (31.4) | 0.54[b] | 18 (26.5) | 21 (30.9) | 0.70[b] |
| Administration of furosemide, n (%) | 3 (4.4) | 3 (1.4) | 0.16[c] | 3 (4.4) | 1 (1.5) | 0.62[c] |
| Estimated blood loss, ml/kg (Median (IQR)) | 1.1 (0.6, 1.7) | 1.4 (0.8, 2.1) | 0.07[a] | 1.1 (0.6, 1.7) | 0.9 (0.6, 1.7) | 0.82[a] |
| Transfusion, n (%) | | | | | | |
| Blood | 14 (20.6) | 38 (18.4) | 0.82[b] | 14 (20.6) | 10 (14.7) | 0.50[b] |
| Plasma | 4 (5.9) | 3 (1.4) | 0.07[c] | 4 (5.9) | 3 (4.4) | 1.00[c] |
| Fluid infusion, ml/kg/h (Median (IQR)) | 27.6 (19.8, 35.8) | 26.9 (21.4, 35.1) | 0.96[a] | 27.6 (19.8, 35.8) | 26.5 (21.2, 35.8) | 0.87[a] |
| Urinary output, ml/kg/h (Median (IQR)) | 2.8 (1.4, 4.1) | 2.5 (1.7, 4.0) | 0.87[a] | 2.8 (1.4, 4.1) | 2.5 (1.4, 4.0) | 0.56[a] |

Notes.

*$P < 0.05$.

[a] $P$ value on a Mann–Whitney U-test.

[b] $P$ value on a chi-squared test.

[c] $P$ value on a Fisher exact test.

the potential confounders, which was one of the strengths of our work. Finally, no significant differences were observed between the two cohorts, either on primary outcomes or secondary outcomes.

As early as 2001, the "off-hour effect" was observed in patients admitted to the emergency department (*Bell & Redelmeier, 2001*). A total of 3,789,917 admissions were analyzed, and the authors concluded that weekend admissions were associated with significantly higher in-hospital mortality rates than were weekday admissions among patients with ruptured abdominal aortic aneurysms, acute epiglottitis, and pulmonary embolism (*Bell & Redelmeier, 2001*). Subsequently, a meta-analysis of 140 identified articles presented that off-hour admission was associated with increased mortality for 28 diverse patient groups (*Zhou et al., 2016*). In addition, a series of studies discussed the "off-hour effect" in patients receiving a surgical intervention, but most focused on adults (*Grandhi et al., 2021*; *Canal et al., 2020*). To our knowledge, no available studies examined the "off-hour effect" in neonates who underwent emergency gastrointestinal surgery. The concept of the "off-hour effect" originated from the fact that the medical service is insufficient, staff shortage, unavailability of some specialized laboratory tests and examinations, and inadequate transportation of employees during off-hours. Those problems seemed more complicated in neonates in many aspects. For neonates, about 5% of patients had acute abdomen (*Rocha et al., 2009*). Of them, 47.2% of cases were born prematurely, and surgical treatments were needed in 68.7% (*Rocha et al., 2009*), indicating some patients could

**Table 3  Primary and secondary outcomes.**

| Variables | Before propensity score matching | | | After propensity score matching | | |
|---|---|---|---|---|---|---|
| | On-hour (n = 68) | Off-hour (n = 207) | P value | On-hour (n = 68) | Off-hour (n = 68) | Adjusted P value |
| **24-hour mortality, n%** | 10 (14.7) | 20 (9.7) | 0.35[b] | 10 (14.7) | 9 (13.2) | 1.00[c] |
| Diaphragmatic hernia | 1 (10.0) | 0 (0.0) | | 1 (10.0) | 0 (0.0) | |
| Gastric perforation | 1 (10.0) | 0 (0.0) | | 1 (10.0) | 0 (0.0) | |
| Intestinal atresia | 1 (10.0) | 2 (10.0) | 0.41[c] | 1 (10.0) | 1 (11.1) | 1.00[c] |
| Intestinal malrotation | 1 (10.0) | 2 (10.0) | | 1 (10.0) | 2 (22.2) | |
| NEC | 6 (60.0) | 16 (80.0) | | 6 (60.0) | 6 (66.7) | |
| **In-hospital mortality, n%** | 12 (17.7) | 25 (12.1) | 0.34[b] | 12 (17.7) | 10 (17.6) | 0.32[b] |
| Congenital anorectal malformation | 0 (0.0) | 1 (4.0) | | 0 (0.0) | 0 (0.0) | |
| Diaphragmatic hernia | 1 (8.3) | 0 (0.0) | | 1 (8.3) | 0 (0.0) | |
| Gastric perforation | 1 (8.3) | 0 (0.0) | 0.43[c] | 1 (8.3) | 0 (0.0) | 1.00[c] |
| Intestinal atresia | 2 (16.7) | 3 (12.0) | | 2 (16.7) | 1 (10.0) | |
| Intestinal malrotation | 1 (8.3) | 2 (8.0) | | 1 (8.3) | 2 (20.0) | |
| NEC | 7 (58.3) | 19 (76.0) | | 7 (58.3) | 7 (70.0) | |
| **Actual PLOS, days (Median (IQR))** | 16.0 (11.0, 25.5) | 18.0 (13.5, 25.5) | 0.18[a] | 16.0 (11.0, 25.5) | 16.0 (11.8, 21.5) | 0.69[a] |
| **Postoperative Hemoglobin, g/dl (Median (IQR))** | 120.0 (100.0, 140.0) | 126.0 (106.0 150.0) | 0.12[a] | 122 ± 36.15 | 130.8 ± 40.2 | 0.18[d] |
| **The incidence of unplanned re-operation, n%** | 11 (16.2) | 26 (12.6) | 0.58[b] | 12 (17.6) | 10 (14.7) | 0.82[b] |

Notes.

*$P < 0.05$.

[a] $P$ value on a Mann–Whitney U-test.

[b] $P$ value on a chi-squared test.

[c] $P$ value on a Fisher exact test.

[d] $P$ value on a $t$-test.

be managed conservatively. Therefore, when neonates need surgical intervention, their condition may be more serious than that receiving conservative treatment. Obviously, for critically ill neonates, to achieve the best outcomes, a multidisciplinary team of surgeons, anesthesiologists, neonatologists, and nurses is required, which is challenging in most institutions due to the specialists' absence during off-hours. Contrasting with the aforementioned studies, a study regarding the impact of birth time in preterm neonates revealed that off-hour births were not associated with increased neonatal mortality (*Yang et al., 2021*). In China, relying on the development of real-time video in recent years, a consultation by the multidisciplinary team could be conducted online, regardless of the time and location, which might guarantee to provide of high-quality medical care for neonates.

We found that the major causes of urgent surgical intervention in neonates were NEC, congenital anorectal malformation, intestinal atresia, and gastric perforation, which was similar to that reported by *Rocha et al. (2009)*. Patients with acute abdomen are referred to them with abdominal symptoms that might rapidly get worse and therefore require immediate treatment. However, even among the patients with acute abdomen, 32.2% of

patients could be treated conservatively (*Rocha et al., 2009*). Once the decision to operate was made, the outcomes were not only based on patients' status but also on surgeons' capability. Theoretically, to maintain pediatric surgeons' capability, a minimum index case per year is required. In 2010, a certified pediatric surgeon performed 9.5 index cases annually in the United States (*Fonkalsrud et al., 2014*), which was lower than the recommendation by *Ravitch & Barton (1974)* that 55 index cases were needed annually to remain competent. The key Ravitch index cases included diaphragmatic hernia, intestinal atresia, imperforate anus, and gastroschisis, which had overlaps with our cases (*Ravitch & Barton, 1974*). About 32.3% of procedures related to pediatric patients were performed by an adult general surgeon (*Purcell et al., 2021*). The possibility of an inexperienced surgeon might contribute to worse outcomes for patients who received surgical intervention during off-hours. However, in the current study, we found that both 24-hour and in-hospital mortality were equivalent for all neonates who accepted emergency surgical intervention, regardless of the surgical time, thus concluding that the quality of medical care did not compromise during the off-hours in our institution. We believed that the following factors could contribute to the homogeneous management between on-hours and off-hours. First, our institution is located in the southwest region of China, which has a population of 20 million. This is one of the largest pediatric hospitals in the local area, which performs over 8,000 pediatric operations per year. According to preliminary investigations, a pediatrician completes more than 200 operations each year in our hospital, indicating the surgeons had abundant experience in dealing with a complicated problem in pediatrics. Next, the on-call pediatric surgeon has more than 3 years of experience in pediatric surgery, and a senior attending is on standby during off-hours, which may be responsible for the high-quality management. Last, at least two anesthesiologists with >2 years' pediatric anesthesia experience were demanded to be present for any neonate's surgery was demanded, which was a standard protocol, even though this led to a higher cost of human resources. Moreover, the neonatal team's proficiency and experience play a crucial role during the health care. A recent study involving 572 premature infants from Turkey reported that there were no significant differences in both mortality and morbidity between on- and off-duty hours. The authors believe that this result is ensured by neonatal team's provision of standardized and experienced care (*Akin & Cakir, 2024*). Undoubtedly, the results from the current study suggest that the neonatal team in our hospital has sufficient experience, and their capabilities remain uncompromised, whether during on hours or off hours.

## Limitations

There are several limitations. First, this is a single-center study. As mentioned above, our hospital is one of the largest pediatric hospitals in the local region, and the surgeons' competency and neonatal teams' experience determine the outcomes of patients. It is important to note that the pediatric surgeons and neonatologists who worked in our institution are more experienced in dealing with neonatal acute abdomen than those in other institutions. Therefore, our results may not be generalized to other institutions, especially small hospitals or non-pediatric hospitals. In the less-resourced settings, a more detailed evaluation of patients' conditions and the treatment capacity of medical

institutions is required, and, if necessary, the patients should be transferred to the hospital with better treatment capacity. Second, the length from the surgical decision to the start of surgery is important for assessing service delays. However, this information cannot be extracted precisely because the surgeon may inform team members (*i.e.,* anesthesiologists and surgical nurses) orally in an urgent situation and then provide written information. Next, it is noted that premature infants are more fragile and prone to develop poor outcomes. This group is special, which requires not only surgical intervention but also the more experienced management of the entire hospital team. However, the sample size of this group is small in the retrospective study. Although 128 patients were identified as premature infants, only 20 patients were extracted after propensity score matching, which could not achieve a confident result. However, we are collecting and analyzing the data continuously. We believe that we will report the results in the near future. Last, selection bias is inevitable because of the retrospective design. The data are collected from electronic medical records, and the accuracy of the original information determines the final result. However, to date, there is no available method for assessing the quality of raw data. In the future, to overcome the bias related to the presence of only one surgical team, a multi-center, prospective study is needed to confirm the results.

## CONCLUSIONS

In this retrospective study with neonates who underwent emergency gastrointestinal surgery, after controlling for age, weight, gestation weeks, and ASA status, the quality of medical care was not compromised during off-hours.

### Funding

This research was funded by the Science & Technology Department of Sichuan Province, grant number 2023NSFSC1626. The funders had no role in study design, data collection and analysis, decision to publish, or preparation of the manuscript.

### Grant Disclosures

The following grant information was disclosed by the author:
Science & Technology Department of Sichuan Province: 2023NSFSC1626.

### Competing Interests

The authors declare there are no competing interests.

### Author Contributions

- Yu Cui conceived and designed the experiments, performed the experiments, analyzed the data, prepared figures and/or tables, authored or reviewed drafts of the article, and approved the final draft.

## Human Ethics

The following information was supplied relating to ethical approvals (i.e., approving body and any reference numbers):

Institutional Review Board (or Ethics Committee) of Chengdu Women And Children Central Hospital (No. B2021(27)).

## Data Availability

The raw data are available in the Supplemental File.

## Supplemental Information

Supplemental information for this article can be found online at http://dx.doi.org/10.7717/peerj.19468#supplemental-information.

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
