# Peer review of "Investigating the “off-hour effect” on outcomes of neonates undergoing emergency gastrointestinal surgery"

_PeerJ, doi:10.7717/peerj.19468_

## Round 0.1 · original submission · Major Revisions

Please respond to the reviews in an appropriate revision.

Reviewer 1 ·

Basic reporting

General Evaluation:
The article is a retrospective study investigating the relationship between the "off-hour effect" (interventions performed during off-hours) and clinical outcomes in neonates undergoing emergency gastrointestinal surgery. The topic is significant for quality assessment in critical areas such as neonatal surgery. The study is supported by detailed analysis and appropriate statistical methods. However, its single-center design and retrospective nature introduce certain limitations regarding the generalizability and objectivity of the findings.
Abstract
Evaluation:
The abstract provides a clear summary of the study's objectives, methods, and findings. However, it could better emphasize the clinical implications of the findings and how they challenge or confirm existing literature. The last sentence is strong but could be expanded to briefly mention how these results might influence clinical practice or policy.
Introduction
Evaluation:
The introduction effectively sets the context for the study, discussing the importance of neonatal gastrointestinal disorders and the concept of the "off-hour effect." However:
• It could provide a more detailed explanation of why the "off-hour effect" might occur, such as staff shortages or differences in expertise.
• The following two articles should also be referenced in the Introduction and Discussion sections. Since this study focuses on neonates, references to neonatal studies should be prioritized over pediatric intensive care studies.
o Toyokawa S, Hasegawa J, Ikenoue T et al. Weekend and off-hour effects on the incidence of cerebral palsy: contribution of consolidated perinatal care. Environ Health Prev Med 2020;25:52. https://doi.org/10.1186/s12199-020-00889-y.
o Akin, M. S., & Cakir, U. (2024). Evaluation of the relationship between day or night birth time and morbidities and mortality in premature infants less than 32 weeks in a Turkish NICU. Journal of Tropical Pediatrics, 70(6), fmae049.

• Including a clearer statement of the study's hypothesis and its novelty compared to prior research would strengthen this section.

Experimental design

Methods
Evaluation:
The methods section is thorough and well-structured. It outlines patient selection, data collection, and statistical analysis clearly. However:
• The propensity score matching (PSM) process is described well, but the rationale for selecting specific covariates could be elaborated.
• When defining off-hour days, it should be explicitly stated whether national holidays, such as national celebrations or public holidays, are considered and how these days were categorized.
• Additionally, considering the timeline of patient admissions between July 2018 and October 2021, the impact of the pandemic period should also be addressed. Did the concepts of on-hour and off-hour remain the same during the pandemic?

Validity of the findings

Results
Evaluation:
The results section is detailed and presents findings in a structured manner. Key points:
• The analysis of primary and secondary outcomes is clear but could benefit from a more explicit focus on clinical significance.
• The presence of 8 appendicitis cases in neonates is intriguing. The accuracy of the diagnosis in these cases should be verified. If any doubts exist, it is recommended to exclude these cases from the study.
• I am curious about the gestational ages and birth weights of the NEC cases. The results indicate statistical differences in birth weight and gestational age. Premature infants should be analyzed as a subgroup, and the findings should be presented either as a table or, if not in a table, as text in the results section. This is because premature infants are the patient group most likely to be negatively affected by the conditions during off-hours. This group, which requires not only surgical intervention but also the more experienced management of the entire hospital team, consists predominantly of premature infants.

Additional comments

Discussion
Evaluation:
The discussion interprets the results well and situates them within the existing literature. Suggestions:
• Expand on potential mechanisms behind the lack of an off-hour effect, considering institutional practices or team expertise.
• The section regarding premature infants should be included in the discussion. It appears that this is the primary affected patient group, which is a logical conclusion.
• In discussing the results, it should be emphasized that not only the surgical team's competence but also the neonatal team's proficiency plays a crucial role and should be addressed in the discussion.
Limitations
Evaluation:
The limitations section appropriately acknowledges the retrospective design, single-center nature, and data constraints. However:
• It could suggest more specific future research directions, such as multi-center studies or prospective designs.
• Discussing how the findings might vary in less-resourced settings would strengthen this section.
• Providing information solely about the surgical team is a limitation. It should be noted that the entire neonatal team (neonatologists, neonatal nurses, etc.) could influence the process and have an impact on mortality and morbidity outcomes.

Reviewer 2 ·

Basic reporting

The study focus on an interesting and underexplored question in the pediatric surgical field. A key strength is the large sample size, despite the study being monocentric and retrospective. The introduction is clear, and although the amount of purely pediatric literature cited is limited, the selected articles are relevant and summarized concisely. The structure of the article is fundamentally correct, and the figures are easy to understand. I believe that the main weakness of the article lies in the level of English, which is only just adequate and may limit the accessibility and comprehension of the text. Some examples of passages with errors include line 34 (reason, not reasons), 53 ("the" is pleonastic), 99 (were obtained, not was obtained), and 135 (needed, not need). In other cases, while the English is fundamentally correct, the phrasing makes it difficult to understand (see lines 41, 46, and 193-198, among others). It should also be noted that in the abstract, the acronym PLOS is first used without explanation (line 21), which is provided only the second time (line 25).
Regarding the figures, in figure 1, at the third level, the central rectangle should read "weekdays 5:01 PM," not AM, and it would be correct to link the box to the” included cases” one, as done for the side panels.

Experimental design

The study objectives are simple and clearly stated; the data analysis is thorough, the tables are easy to understand, and the results are discussed in sufficient detail. In the materials and methods section, the explanation of the diagnostic and surgical approach, as well as the comparison between daytime and off-hour cases could be clearer, as it is somehow convoluted and it lacks fluidity. The study’s limitations are well-developed, and I really appreciated them for the researcher's insight on their own work.

Validity of the findings

Given the limited literature on this topic, even a monocentric experience is of scientific interest. It would be highly beneficial to expand the investigation by developing a national multicenter study or even making it international, in order to overcome the bias related to the presence of only one surgical team.

---

## Round 0.2 · accepted · Accept

Thank you for submitting your revised manuscript entitled "Investigating the 'off-hour effect' on outcomes of neonates undergoing emergency gastrointestinal surgery" to PeerJ.

We appreciate the careful and thoughtful revisions you have made in response to the reviewers' comments. The manuscript has been significantly improved and now meets the editorial and scientific standards required for publication.

I am pleased to inform you that your manuscript has been accepted for publication in PeerJ. Congratulations!

Reviewer 1 ·

Basic reporting

Clear and unambiguous, professional English used throughout.

Experimental design

Original primary research within Aims and Scope of the journal.

Validity of the findings

Impact and novelty not assessed. Meaningful replication encouraged where rationale & benefit to literature is clearly stated.